# Methylation Biomarkers of Lung Cancer Risk: A Systematic Review and Meta-Analysis

**DOI:** 10.3390/cancers17040690

**Published:** 2025-02-18

**Authors:** Jacopo Dolcini, Manuela Chiavarini, Giorgio Firmani, Kasey J. M. Brennan, Andres Cardenas, Andrea A. Baccarelli, Pamela Barbadoro

**Affiliations:** 1Department of Biomedical Sciences and Public Health, Section of Hygiene, Preventive Medicine and Public Health, Polytechnic University of the Marche Region, 60126 Ancona, Italy; 2Department of Health Sciences, University of Florence, 50134 Florence, Italy; 3Department of Environmental Health, Harvard T.H. Chan School of Public Health, Boston, MA 02115, USA; 4Department of Epidemiology and Population Health, Stanford University, Palo Alto, CA 94305, USA

**Keywords:** lung cancer, DNA methylation, blood, meta-analysis, systematic review, epigenetic age, public health

## Abstract

Lung cancer (LC) is among the leading causes of cancer deaths worldwide, and understanding factors that increase risk could improve prevention and early detection efforts. Several indicators, such as epigenetic age and others that analyze methylation levels, have been shown to predict aging and disease risk in relation to LC. Still, there is no clear consensus on how to implement these measures for clinical use or prevention strategies. This review and meta-analysis examines the link between DNAm in blood and the risk of developing LC. By combining data from existing studies, we found that higher levels of DNAm are associated with a greater risk of LC. These results suggest that DNAm and related indicators could be used as biomarkers to identify individuals at higher risk of LC. Further research is needed to clarify how DNAm patterns could be used in LC screening and to understand other factors that might influence this relationship.

## 1. Introduction

According to the International Agency for Research on Cancer, with an estimated 2.48 million new cases and 1.81 million deaths per year in 2022 [1], lung cancer (LC) is one of the most frequently diagnosed cancers and the leading cause of cancer-related deaths worldwide. Not only is LC highly prevalent, but it has also shown an increased trend over time [2,3]. Public health campaigns to reduce smoking have contributed to a partial but not sufficient decrease in incidence and mortality in several countries [4,5]. Significative improvements have been made in terms of the general understanding of disease biology, risk factors, the application of predictive biomarkers, and refinements in treatment that have led to remarkable progress for the health outcomes of many patients [6,7]. However, prevention strategies and screening still remain limited, as LC survival has mostly improved due to new treatments rather than screening [3]. Preventive chest X-rays with or without sputum cytology did not show significant results in terms of patient outcomes [8]. Some benefits have been shown through screening focusing on high-risk groups, identified based on exposure to high levels of cigarette smoking [9,10], and using more advanced approaches like low-dose CT [10]. However, these technologies have higher costs and lower availability, especially in low- and middle-income countries. It is well known that LC can also occur in subjects without a history of high-risk exposure, such as smoking [11]. However, these populations are often not effectively reached by conventional prevention campaigns. Therefore, new minimally invasive biomarkers involving blood or sputum sampling have gained interest [12]. From these specimens, biological information can be obtained through epigenetic analyses involving DNA methylation (DNAm).

Over the past decade, there has been growing interest in using epigenetic changes as markers for early cancer detection [13]. Epigenetic dysregulation, including in DNAm, is associated with many tumor types, including LC, and chemotherapy resistance [14]. Based on the hypothesis that DNA from apoptotic cancer cells can be detected in serum and plasma, DNAm changes in several target genes have been evaluated in blood samples of patients with LC. Additionally, some evidence suggests that certain aberrant DNAm in the blood might reflect pathological changes in target tissues that cannot otherwise be identified easily or safely [15]. Other studies that focused on a specific set of genes, such as *RASSF1A*, *CDKN2A*, *RARb*, *CDH13*, *FHIT*, and *BLU*, showed an accordance of methylation levels between blood samples and tumor samples [16]. Following increasing interest in DNAm and other epigenetic mechanisms, in recent years many studies have focused on the development of biomarkers that look at methylation and quantitatively combine DNAm levels at tens to hundreds of genomic locations into composite methylation-based age predictors. These measures often exhibit extremely high correlations with chronological age levels but can assess inter-individual and/or inter-tissue variability in the rate of aging, possibly capturing differences between epigenetic or biological aging and chronological aging [17]. These differences can be related to differential susceptibility to death [18,19] or diseases [20,21,22]. Looking at global DNAm modifications has led to a broader understanding of LC tumorigenesis and has shown promising candidates for early LC detection or for the selection of high-risk subjects with further specific screening methods [23,24]. This is why also epigenetics clocks that are usually built on global methylation analyses have been found to be associated with increased cancer risk [21,25,26], including LC [27]. It must be noted that, in contrast, some other studies encountered a weak association or a lack of one [28,29,30], showing a lack of consensus probably due to several factors such as reverse causation and residual confounding.

## 2. Materials and Methods

The present meta-analysis was conducted following the MOOSE (Meta-analysis Of Observational Studies in Epidemiology) guidelines [31] and PRISMA (Preferred Reporting Items for Systematic reviews and Meta-Analyses) statement [32].

The study protocol was registered in the International Prospective Register of Systematic Reviews (www.crd.york.ac.uk/PROSPERO/, registration No: CRD42024516565) (accessed on 22 February 2024).

### 2.1. Search Strategy and Data Sources

We carried out a systematic literature search up to 13 February 2024 through the PubMed http://www.ncbi.nlm.nih.gov/pubmed/ (accessed on 14 February 2024), Web of Science http://wokinfo.com/ (accessed on 14 February 2024), and Scopus https://www.scopus.com/ (accessed on 14 February 2024) databases to identify original articles on the association between changes in DNAm and LC. The following key words were used: (DNA methylation) AND (lung cancer risk). The different associations of keywords combined with Boolean operators used for each database are shown in Appendix A. No publication date limitation was applied, but due to translation restrictions, only English-language studies were eligible. In addition, to identify additional relevant publications, we manually examined the reference lists of the included articles and recent relevant reviews.

### 2.2. Eligibility Criteria

Of the selected articles, only those that met the following criteria were included: (i) evaluated the relationship between changes in DNAm and LC; (ii) used a case–control, prospective, or cross-sectional study design; (iii) reported odds ratio (OR), relative risk (RR), or hazard ratio (HR), estimated with 95% confidence intervals (CIs). Studies that aggregated LC cases with other types of cancer were excluded. For each potentially included study, two investigators independently carried out the selection, data extraction, and quality assessment. Disagreements were resolved by discussion or in consultation with a third author. Although useful to have background information, reviews and meta-analyses were excluded. No studies were excluded for weakness of design or data quality.

### 2.3. Data Extraction and Quality Assessment

For each selected study, we extracted the following information: first author’s last name, year of publication, country, study design, sample size, population characteristics (sex, age, race, BMI, smoking status), duration of follow-up for cohort studies, risk estimates with 95% CIs, type of LC, tissue type, DNAm measure, and confounding factor adjustment. The effect estimates represent a relative change in DNAm rather than an absolute increase measured in specific units, which may be associated with aging as measured by epigenetic clocks, smoking, or the methylation risk score. Our results are derived from the analysis of epigenetic clocks, which are regression models constructed in studies comparing DNA methylation levels between populations diagnosed with lung cancer and control groups without lung cancer. While these studies used diagnosis as a reference for model construction, the resulting regression model is capable of assessing risk based on given methylation levels. When multiple estimates were reported in the article, we extracted those adjusted for the most confounding factors. The Newcastle–Ottawa Scale (NOS) was used to assess the quality of the literature for cohort and case–control studies using a 9-star system, as shown in Appendix A [33]. The full score was 9 and a total score ≥ 7 was used to indicate a high-quality study [32].

### 2.4. Statistical Analysis

The overall effect-size statistic estimate was carried out considering the risk of LC associated with the highest versus the lowest level of changes in DNAm. The risk values of the multivariable models were selected by considering those that took into account the greatest number of potential confounding variables. The relative risk and hazard ratio were taken as an approximation to the OR, and the meta-analysis was performed as if all types of ratios were ORs. Due to the high heterogeneity, a random effects model and inverse variance weighting methods were used to calculate the sum of the OR and the 95% confidence intervals. An effect was considered statistically significant when a two-tailed *p* < 0.05 was obtained. We further aimed to quantify the effects of DNAm on risk of LC using a dose–response meta-analysis with random effects models [34,35]. To evaluate the dose–response relationship between exposure and outcome, data were extracted from studies that reported results by exposure quartiles (Q1, Q2, Q3, Q4). The outcome variable considered was the odds ratio (OR) for each quartile, with corresponding lower (LL) and upper (UL) 95% confidence intervals. For the reference quartile (Q1), the odds ratio was set to 1 (log(OR) = 0). For the subsequent quartiles (Q2, Q3, Q4), the logarithm of the OR was calculated to provide a linear representation of the dose–response relationship. Variance was calculated using the confidence intervals, following a well-established procedure to estimate uncertainty in meta-analyses [36,37]. Exposure was standardized on a continuous scale, assigning values of 0, 1, 2, and 3 to quartiles Q1, Q2, Q3, and Q4, respectively. This approach is commonly used to explore dose–response relationships in epidemiological data and allows the effect of increasing exposure to be modeled as a continuous variable [35]. A random effects meta-regression was conducted using the metafor package in R [38], employing the restricted maximum likelihood method (REML). The independent variable in the regression was the standardized exposure, with the goal of estimating the incremental effect of exposure in the higher quartiles compared to Q1. The random effects model was chosen to account for heterogeneity among the included studies [39]. Predicted ORs for each quartile were obtained from the meta-regression results. The OR estimates for the subsequent quartiles were compared to Q1 (OR = 1) to assess the dose–response relationship between exposure and outcome. Additionally, confidence intervals were constructed for each prediction. A dose–response plot was created to visualize the relationship between standardized exposure (quartiles) and odds ratios. Quartile Q1 was used as the reference (OR = 1), and the subsequent quartiles were plotted relative to Q1, clearly representing how the odds ratio changes with increasing exposure. This procedure is commonly used to explore and present the dose–risk relationship in epidemiological studies [34].

The chi-square-based Cochran’s Q statistic and the I^2^ statistic were used to evaluate the heterogeneity in the results across the studies [40]. The I^2^ statistic yields results ranging from 0% to 100% (I^2^ = 0–25%, no heterogeneity; I^2^ = 25–50%, moderate heterogeneity; I^2^ = 50–75%, large heterogeneity; and I^2^ = 75–100%, extreme heterogeneity) [41]. The results of the meta-analysis may be biased if the probability of publication is dependent on the study results.

We used the methods of Begg and Mazumdar [42] and Egger et al. [43] to detect publication bias. Both methods tested for funnel plot asymmetry, with the former being based on the rank correlation between the effect estimates and their sampling variances and the latter on the linear regression of a standard normal deviate and its precision. If a potential bias was detected, we conducted a further sensitivity analysis to assess the strength of the combined effect estimates and the possible influence of the bias, and to have the bias corrected. We also conducted a sensitivity analysis to investigate the influence of a single study on the overall risk estimate by omitting one study in each turn. We considered the funnel plot to be asymmetrical if the intercept of Egger’s regression line deviated from zero, with a *p*-value < 0.05. The analyses were performed using the ProMeta 3 statistical program and the calculations on the data extracted from the original papers were performed using STATA 13.3.

## 3. Results

### 3.1. Study Selection

The literature search revealed 582 studies from the PubMed database, 955 from Web of Science, and 829 from Scopus. After removing duplicates (*n* = 794), we identified 1572 records screened for title and abstract revision. Among these, 1546 articles were excluded (reviews, pooled or meta-analyses, commentaries, and case studies) (Figure 1).

Therefore, 26 studies were subjected to full-text revision. We identified one additional study through the reference lists of both the selected articles and recent relevant reviews. Subsequently, 16 articles were excluded because they did not meet the inclusion criteria, since they were studies on single-gene methylation, on animal models, or focused on the promoter methylation of specific genes (Appendix A). We excluded articles that addressed these points from this systematic review and meta-analysis for the following reasons: methylation of a single gene was not the primary focus of methylation regression models; this systematic review and meta-analysis focused on human studies and not animal models; methylation as an epigenetic mechanism extends beyond the promoter region; and the NOS is not applicable to randomized studies such as Mendelian randomization. At the end of the selection process, 11 studies were eligible for inclusion in the systematic review and meta-analysis [27,28,44,45,46,47,48,49,50,51,52].

### 3.2. Study Characteristics and Quality Assessment

The general characteristics of the 11 studies evaluating the association between DNAm and LC risk, as well as the relative populations included in the meta-analysis, are shown, respectively, in Table 1.

Studies were conducted in the United States of America [45,48,52]; Australia [27,44,46,49,51]; Germany [47,50]; Italy [44]; Scotland [28]; and Norway [44]. There was one case–control study [45], eight nested case–control studies [27,44,46,47,48,51,52], and two cohort studies [28,50]. Studies were published from 2015 to 2023, and all of them were conducted in adults aged ≥50 years old.

The health outcome investigated was the risk incidence of LC. The study-specific quality scores of selected studies are shown in the last column on the right of Table 1. The quality scores range from 7 to 9 (median: 8; mean: 8) in Table 1.

### 3.3. Meta-Analysis of DNAm

Regarding DNAm (Table 2 and Figure 2), we found an overall increased risk associated with greater values of DNAm (OR = 1.24 95% C.I. 1.10–1.39) that showed a stronger association and statistical significance when focusing only on cohort studies (OR = 1.61, 95% C.I. 1.36–1.90). Looking at stratification according to smoking status, we found a statically significant risk increase both for former smokers (OR = 1.58; 95% C.I. 1.41–1.77) and current smokers (OR = 1.60; 95% C.I. 1.43–1.79). A stratification based on follow-up time period revealed a statistically significant increased likelihood of LC risk with greater DNAm with a follow-up of ≤ 5 years (OR = 1.46; 95% C.I. 1.08–1.98) and 5–10 years (OR = 1.20; 95% C.I. 1.06–1.36). We then stratified the several different types of epigenic clocks and indicators according to similar characteristics (if they measured accelerated age (AA) or intrinsic epigenetic age acceleration (IEAA)), but we did not observe any associations for grouped indicators. All DNAm measure definitions, which are taken in consideration, are described in Appendix A. Regarding the single epigenetic clock (Table 2), we observed a statistical association only for the epigenetic age acceleration of GrimAge (OR = 1.97; 95% C.I. 1.57–2.47). Because of the small amount of data, no further stratification according to other variables such as body mass index (BMI), age, or race was possible. Concerning heterogeneity, the analysis revealed significant heterogeneity across the studies. In the combined analysis (ALL), the heterogeneity was high (Q = 163.95, I^2^ = 93.90%, *p* < 0.001), indicating substantial variability between studies. This high level of heterogeneity was consistent across various subgroup analyses, including case–control studies (Q = 350, I^2^ = 70.57%, *p* < 0.001) and cohort studies (Q = 7.01, I^2^ = 14.42%, *p* = 0.32), though the latter showed lower heterogeneity. For follow-up periods, studies with ≤5 years showed moderate heterogeneity (Q = 4.92, I^2^ = 59.36%, *p* = 0.09), studies with a 5–10-year follow-up showed no heterogeneity (Q = 1.27, I^2^ = 0.00%, *p* = 0.53), and studies with a follow-up of >10 years showed high heterogeneity (Q = 12.97, I^2^ = 76.87%, *p* = 0.005).

### 3.4. Sensitivity Analysis of DNAm

Sensitivity analyses investigating the influence of a single study on the LC risk estimates suggested that these were not substantially driven by any single study. Indeed, the LC risk estimates ranged from 1.07 (95% CI 1.01–1.13, *p* = 0.025), omitting the study of Yu et al. [47], to 1.17 (95% CI 1.10–1.24, *p* < 0.001), omitting the study of Michaud et al. [52].

### 3.5. Publication Bias of DNAm

Publication bias was detected for ALL and cohort studies with Egger’s method and for current smoking status with Begg’s method (Table 2 and Figure 3).

### 3.6. Dose–Response

Three articles were identified for dose–response analysis [47,50,52]. The other studies were excluded because they did not include division by quartiles [27,28,44,45,46,48,49,51].

The dose–response relationship between methylation levels, expressed as standardized quartiles, and LC risk is presented in Figure 4. The *x*-axis represents the standardized exposure levels corresponding to methylation quartiles Q1 (reference), Q2, Q3, and Q4, while the *y*-axis represents the odds ratio (OR) for LC risk on a linear scale. The red dashed horizontal line at OR = 1 marks the reference category (Q1).

The black solid line in the figure illustrates a slight increase in ORs across the quartiles, indicating a potential rise in LC risk with increasing methylation levels. The blue dashed lines represent the 95% confidence intervals, which widen at higher quartiles, suggesting increased uncertainty in the estimates for the upper methylation quartiles.

The mixed effects meta-regression model indicated that individuals in the lowest methylation quartile (Q1) had reduced odds of LC, with an OR of 0.77 (95% CI: 0.63 to 0.94). This suggests that individuals in Q1 have about 23% lower odds of developing LC compared to the higher quartiles.

For each one-unit increase in standardized exposure (i.e., moving from one quartile to the next), the odds ratio for LC was 1.03 (95% CI: 0.95 to 1.13). This indicates a slight, but non-significant, increase in LC risk with higher methylation quartiles.

The wide confidence intervals in the higher quartiles indicate some uncertainty in the risk estimates, and no conclusive dose–response pattern was observed.

## 4. Discussion

Our results are consistent with the growing body of evidence in the literature that links changes in DNAm patterns or biomarkers with LC risk [53]. These methylation changes have been deeply investigated, with a particular focus on specific regions, such as the promoter region of the CDKN2A gene (Cyclin-Dependent Kinase Inhibitor 2A), which has been identified as one of the first potential new biomarkers for the early detection of LC [52,54]. In addition, a more comprehensive approach has been taken through modern measures, such as epigenetic clocks, which can also be related to LC [27,45], even if some studies did not find a strong association [46,52]. We found a strong significant association for the overall effect and for several types of stratification, including smoking status (Table 2). This is in agreement with previous studies that found a stable epigenetic signature of cigarette smoking, both in current and past smokers [55,56,57]. Interestingly, the follow-up period (Table 2) seems to represent an important discrimination in terms of DNAm biomarkers and LC risk: for time periods ≤ 5 years and from 5 to 10 years of observation, the risk increases in a statistically significant manner with a higher risk in the ≤5 years class. Then, the risk decreases and is no longer statistically significant after ten years. It can be hypothesized that epigenetic modifications have a much higher impact on LC risk within the first five years of their occurrence, and their effect gradually decreases over time. This is in accordance with some studies that observed that some methylation changes induced by smoking can reverse within 5 years of quitting. Anyway, a large portion of CpG sites remain persistently altered for decades. These findings highlight both the substantial reversibility of smoking-related epigenetic modifications and the potential for long-lasting effects at specific loci [55]. We then stratified our results by grouping several epigenetic clocks that, while built on different premises, share some common characteristics. For example, the Horvath clock and Hannum clock are built by regressing DNAm on chronologic age, but the first is a pan-tissue clock, originally constructed utilizing CpG sites across 51 human cell types and tissues, while the Hannum clock was developed by regressing only peripheral blood samples, and includes 71 CpG sites [58,59]. The PhenoAge clock was created using blood samples, similar to Hannum, but it regresses DNAm states on clinical biomarkers rather than on chronologic age and incorporates 513 CpG sites [21]. Moreover, we created groups considering the papers analyzed in our investigation: the authors adjusted these clocks by regressing DNAm age on chronological age and calculating the difference between the observed chronological age and the fitted DNAm age, obtaining DNAm age acceleration (AA) [51]. Intrinsic epigenetic age acceleration (IEAA) was then derived using the residuals from the linear regression of DNAm age on chronological age, further adjusted for estimated blood cell composition [51]. These adjustments allowed for more comparable indicators. Despite these attempts, only the GrimAge epigenetic age acceleration, calculated on the residuals from regression models of GrimAge estimates on chronological age [27], showed a statistically significant increased risk of LC incidence. These results may be due to the nature of the epigenetic clocks built in a two-stage process, where the first stage involved the inclusion of DNAm-based surrogate biomarkers such as smoking pack-years [60]. The lack of significance for the stratification and grouping of the several different epigenetic measures may be due to both the low number of studies that were eligible for our analyses and differences in the biological elements and aspects that are captured by each of these clocks, even if they share common characteristics, as previously mentioned. Interestingly, the dose–response analysis showed a reduced likelihood of incidence for LC risk for subjects in the first quartile that was statistically significant. While not statistically significant, increasing quartiles of methylation levels showed a trend of increased risk. This increased risk is in agreement with the overall effect we encountered, and the loss of significance may be due to the fact that the dose–response analyses was performed on a reduced number of studies, i.e., those that reported quartiles of methylation levels. From a biological point of view, smoking is known to promote DNA methylation at certain loci, leading to the silencing of tumor suppressor genes and the activation of oncogenes. Over time, these epigenetic alterations can accumulate, potentially increasing the risk of lung cancer even in the absence of a direct dose–response pattern in the available data. This is consistent with epidemiological findings showing that the longer a person smokes, the higher their lung cancer risk.

Furthermore, the complexity of epigenetic regulation means that methylation changes may have non-linear effects on gene expression, making it challenging to detect a straightforward dose–response trend in case–control studies. Other confounding factors, such as inter-individual variability in epigenetic plasticity and the interplay with additional genetic and environmental influences, may further obscure a simple dose–response relationship [55]. Moreover, the heterogeneity of biomarkers may have influenced this result since some are aging clocks while others are biomarkers of smoking and/or methylation risk scores. It is possible that the association is driven primarily by those biomarkers, such as GrimAge, that have been built on smoking habits but end up more strongly associated with smoking effects than smoking itself.

### Strengths and Limitations

To the best of our knowledge, this is the first study to conduct a systematic review and metanalysis of several DNAm measures in relation to the risk of LC incidence, representing a comprehensive, if incomplete, approach to better understanding the relationship of this possible promising biomarker with LC. Moreover, all studies included were of high quality, scoring above 7 on the NOS, ensuring reliability, even if we are aware of their possible limitations, especially in terms of the inter-rater agreement between observers of the NOS [61,62]. All articles included in this analysis were published within the last decade, reflecting the most current research in this field. In addition, all studies included had a considerable time range of follow-up of up to 10 years that allowed the proper stratification of our results. However, several limitations should be acknowledged. The considerable heterogeneity observed across studies can be attributed to differences in several analytic factors affecting the several different types of epigenetic clocks and indicators that were used in the selected studies. Another limitation is the potential impact of intra-tumoral epigenetic heterogeneity, which was not directly assessed in the included studies. The studies included in our analysis did not directly assess this variability at the tumor level, as they relied on blood-derived DNAm measurements rather than intra-tumoral sampling. The reliance on blood-derived DNAm measurements means that variability in tumor subclones could indirectly influence the findings. The release of DNA from apoptotic tumor cells into the bloodstream may reflect diverse tumor subclones, potentially introducing variability in the DNAm patterns detected. To mitigate this, we applied random effects models to account for between-study variability and performed subgroup analyses. Anyway, future research integrating data from both tumor tissues and matched blood samples, as well as single-cell epigenomic technologies, is needed to better understand the role of intra-tumoral heterogeneity in systemic DNAm signatures and their potential as biomarkers.

Additionally, due to limited data availability, further stratification according to BMI, age, socioeconomic status, or other demographic characteristics was not possible.

## 5. Conclusions

This systematic review and meta-analysis suggest that DNAm is a potentially promising biomarker for LC risk prediction. However, more studies are needed to better identify stratification algorithms and strategies to develop specific epigenetic clocks and measures, which must be validated in different populations. The literature shows an increasing number of epigenetic measures, and systematic organization and classification are needed both to reduce heterogeneity among different studies and to enable these tools to be used in clinical practice and prevention campaigns. Public health systems and policymakers should increase efforts to apply evidence-based medicine in the evaluation context of epigenetic tools for LC risk prevention, ensuring that all citizens can access these potentially powerful instruments as reliable and responsible resources for their health.

## Figures and Tables

**Figure 1 cancers-17-00690-f001:**
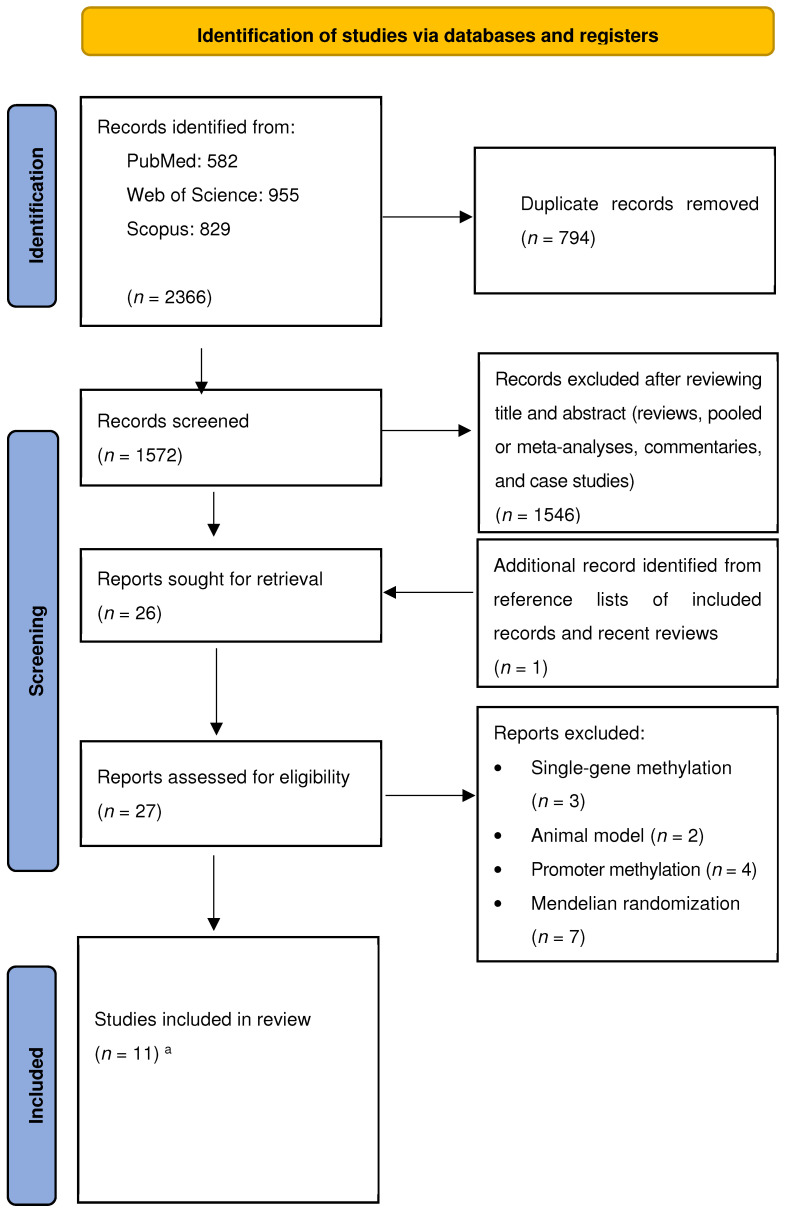
Flow diagram of the systematic literature search on DNAm and LC risk. ^a^ Studies included in review are [27,28,44,45,46,47,48,49,50,51,52].

**Figure 2 cancers-17-00690-f002:**
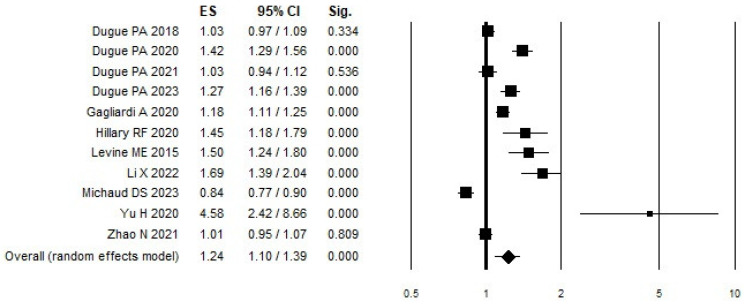
Forest plot of DNAm and risk of LC. References of studies in the figure from top to bottom are following: [27,28,44,45,46,47,48,49,50,51,52].

**Figure 3 cancers-17-00690-f003:**
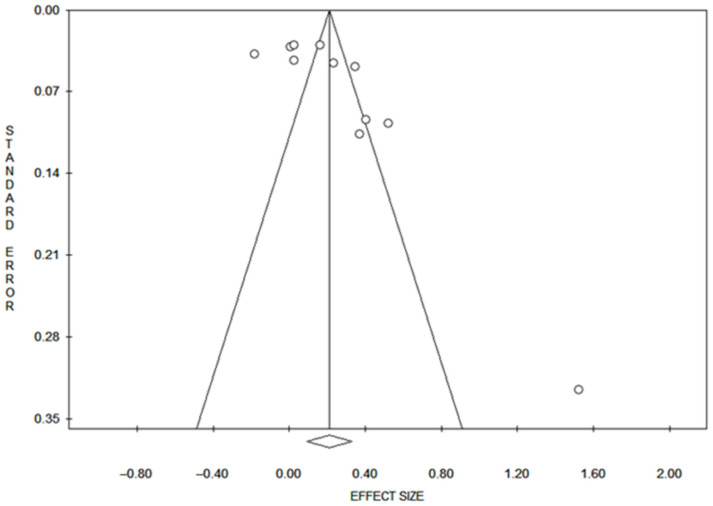
Funnel plot of publication bias of DNAm and risk of LC.

**Figure 4 cancers-17-00690-f004:**
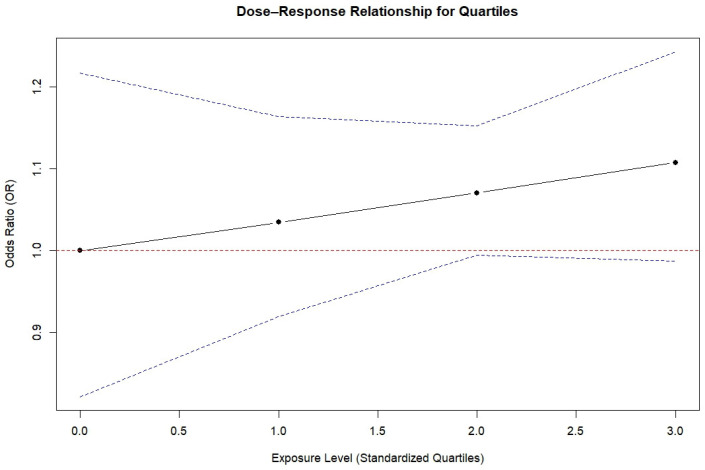
Dose-dependent plots displaying the relation between DNAm and LC risk. The *x*-axis represents the standardized exposure levels corresponding to methylation quartiles Q1 (reference), Q2, Q3, and Q4, while the *y*-axis represents the odds ratio (OR) for LC risk on a linear scale. The red dashed horizontal line at OR = 1 marks the reference category (Q1). The blue lines represent the confidence interval.

**Table 1 cancers-17-00690-t001:** Characteristics of the studies included in the systematic review and meta-analysis of the association between DNAm and LC risk. All studies collected samples from peripheral blood. NA: Not Available.

Author, Year, Reference	Cohort ^1^, Location	Study Design	DNAm Measure ^2^	N	Age (M: Mean, Mdn: Median)	Sex (% Male)	Race (% White)	BMI (%, M: Mean, Mdn: Median [kg/m^2^] < 25)	Type of LC ^3^	Smoking Status (% Never)	Matched or Adjusted Variables	NOS ^4^
Michaud DS, 2023 [52]	CLUE II, USA	Nested case–control	**Epigenetic Age** AA_Hannum, AA_Horvath, AA_Pheno, IEAA_Hannum, IEAA_Horvath, IEAA_Pheno	Cases: 208 Controls: 208	Cases: 58.3 (M) Controls: 55.9 (M)	Cases: 45.7 Controls: 45.7	NA	Cases: 26.0 (M) Controls: 26.2 (M)	LC (NSCLC: 74%)	Cases: 10.6 Controls: 10.6	Batch effects, BMI, smoking predicted years	8
Dugue PA, 2023 [51]	MCCS, Australia	Nested case–control	**SMOKING** Smk-233, Smk-1061. **BMI** BMI-1109, BMI-85. **ALCOHOL CONSUMPTION** Alc-450, Alc-459	Cases: 327 Controls: 327	Cases: 61 (Mdn) Controls: 61 (Mdn)	Cases: 61 Controls: 61	NA	Cases: 27 (M) Controls: 27 (M)	LC	Cases: 46 Controls: 48	Smoking details, physical activity, diet quality, education, SES, alcohol consumption, BMI	9
Li X, 2022 [50]	ESTHER, Germany	Cohort study; FU: 17 years	**Epigenetic Age** AgeAccelPheno, AgeAccelPheno **Methylation Score** MRscore-8CpGs	Cases: 207 Controls: 205	Cases: 63.2 (M) Controls: 62.5 (M)	Cases: 63.8 Controls: 42.9	NA	Cases: 23.7 (%) Controls: 31.2 (%)	LC	Cases: 30.4 Controls: 49.5	Age, sex, leukocyte composition, batch, educational level, smoking status, alcohol consumption, BMI, diabetes status	8
Dugue PA, 2020 [27]	MCCS, Australia	Nested case–control	**Epigenetic Age** PhenoAge, GrimAge	Cases: 327 Controls: 327	Cases: 61 (Mdn) Controls: 61 (Mdn)	Cases: 69.3 Controls: 69.3	NA	Cases: 27 (Mdn) Controls: 27 (Mdn)	LC	Cases: 45.8 Controls: 45.8	Age, sex, country of birth, sample type, smoking information, BMI, height, alcohol consumption, physical activity, dietary quality, socioeconomic status, education	8
Dugue PA, 2021 [49]	MCCS, Australia	Nested case–control	**MATERNAL SMOKING** MS-568, MS-19, MS-15, MS-28, MS-17 **ADULT SMOKING** AS-233, AS-56, AS-1061	Cases: 327 Controls: 327	Cases: 61 (Mdn) Controls: 61 (Mdn)	Cases: 61 Controls: 61	NA	NA	LC	Cases: 48 Controls: 46	Smoking information, alcohol consumption, BMI, physical activity, dietary quality, education, socioeconomic status, height	8
Zhao N, 2021 [48]	CLUE I-II, USA	Nested case–control	mdNLR; CRP Score 1, 2, 3.	Cases: 208 Controls: 208	Cases: 59 (Mdn) Controls: 56 (Mdn)	Cases: 45.7 Controls: 45.7	Cases: 98.6 Controls: 100	NA	LC (all LC and NSCLC)	Cases: 10.6 Controls: 10.6	Age, sex, smoking status, BMI, batch effects, predicted pack-years smoked, cell proportions	9
Hillary RF, 2020 [28]	GS, Scotland	Cohort	DunedinPoAm	Cases: 4450 Controls: 2578	Cases: 51.4 (M) Controls: 50 (M)	Cases: 43.7 Controls: 38.6	NA	Cases: 26.8 (M) Controls: 27.2 (M)	LC	NA	Age, sex, alcohol consumption, BMI, deprivation, education, smoking	8
Yu H, 2020 [47]	ESTHER, Germany	Nested case–control	**Methylation Risk Score** MRS	Cases: 143 Controls: 1460	Cases: 63.7 (M) Controls: 61.8 (M)	Cases: 62.9 Controls: 44.0	NA	NA	LC	Cases: 12.1 Controls: 49.7	Batch, leukocyte composition, age, sex, smoking status, pack-years	9
Gagliardi A, 2020 [44]	EPIC, Italy	Nested case–control	LogSEM model 4; LogSEM model EPIC, LogSEMmodel(TTD) ≤ 5 y, 5–10 y, >10 y	Cases: 556 Controls: 556	Cases: 53.7 (M) Controls: 53.5 (M)	Cases and controls: 31	NA	NA	LC	Cases: 43 Controls: 49	Age, sex smoking, BMI, dietary quality, alcohol intake, physical activity, education, Horvath DNAmAge epigenetic AA, DNAmGrimAge, epigenetic AA	9
	MCCS, Australia	Nested case–control	LogSEM model 4; LogSEM model MCCS, LogSEMmodel(TTD) ≤ 5 y, 5–10 y, > 10 y	Cases: 3482 Controls: 3482	Cases: 59.1 (M) Controls: 58.9 (M)	Cases and controls: 61	NA	NA	LC	Cases: 46 Controls: 48	Age, sex smoking, BMI, dietary quality, alcohol intake, physical activity, education, Horvath DNAmAge epigenetic AA, DNAmGrimAge, epigenetic AA	9
	NOWAC, Norway	Nested case–control	LogSEM model 4; LogSEM model NOWAC, LogSEMmodel(TTD) ≤ 5 y, 5–10 y, > 10 y	Cases: 316 Controls: 316	Cases and Controls: 55.9 (M)	Cases and Controls: 0	NA	NA	LC	Cases: 26 Controls: 38	Age, sex smoking, BMI, dietary quality, alcohol intake, physical activity, education, Horvath DNAmAge epigenetic AA, DNAmGrimAge, epigenetic AA	9
Dugue AP, 2018 [46]	MCCS, Australia	Nested case–control	**Epigenetic Age** AA_Hannum, AA_Horvath, IEAA_Hannum, IEAA_Horvath, EEAA.	Cases: 332 Controls: 332	Cases: 59.5 (M) Controls: 59.4 (M)	Cases and Controls: 64	NA	Cases: 37.0 (%) Controls: 28.0 (%)	LC	Cases and Controls: 12	BMI, smoking, alcohol intake, diet quality, physical activity, socioeconomic status education, age, sex, ethnicity	8
Levine ME, 2015 [45]	WHI, USA	Case–control	**Epigenetic Age** IEAA by age: All ages, 50–59, 60–69, 70+. IEAA by smoking status: Current, Former, Never.	Cases: 43 Controls: 1986	65.34 (M: Cases + Controls)	Cases: 0 Controls: 0	NA	NA	LC	54.4 (M: Cases + Controls)	Age, race/ethnicity, CHD status, pack-years, smoking status	7

^1^ Cohort acronyms: **CLUE (I/II),** Campaign Against Cancer and Stroke; **ESTHER**, Epidemiological Study on the Chances of Cure, Early Detection and Optimized Therapy of Chronic Diseases in the Elderly Population; **MCCS**, Melbourn Collaborative Cohort Study; **EPIC**, European Prospective Investigation into Cancer and Nutrition; **NOWAC**, Norwegian Women and Cancer Study; **WHI**, Women Health Initiative; **GS**, Generation Scotland. ^2^ DNAm measure acronyms: **AA**, age acceleration; **IEAA**, intrinsic epigenetic age acceleration; **EEAA**, extrinsic epigenetic age acceleration; **MRS**, methylation risk score; **LogSEM**, logistic structural equation modeling; **MS**, maternal smoking; **AS**, adult smoking; **MRscore**, methylation risk score, **8CpGs**, eight specific CpG sites; **DunedinPoAm**, Dunedin pace of aging methylation. ^3^ LC acronyms: **LC**, lung cancer; **SCLC**, small-cell lung cancer; **NSCLC**, non-small-cell lung cancer; **LAUD**, lung adenocarcinoma; **LUSC**, lung squamous cell carcinoma. ^4^ Newcastle–Ottawa Scale.

**Table 2 cancers-17-00690-t002:** Results of stratified analysis of the LC risk estimates associated with DNAm.

	Combined Risk Estimate ^a^		Test of Heterogeneity			Publication Bias	
	*N* ^b^	Value (95% CI)	Q	I^2^ %	*p*	*p* (Egger Test)	*p* (Begg Test)
ALL (11 articles)	111	1.24 (1.10–1.39)	163.95	93.90	0.00	0.032	0.07
ALL (Case–control study)	104	1.05 (0.99–1.11)	350	70.57	0.00	0.09	0.14
ALL (Cohort study)	7	1.61 (1.36–1.90)	7.01	14.42	0.32	0.012	0.18

Smoking status							
CURRENT	7	1.60 (1.43–1.79)	2.65	0.00	0.45	0.06	0.042
PAST	7	1.58 (1.42–1.77)	4.11	0.00	0.66	0.09	0.19
Follow-up							
≤5	3	1.46 (1.08–1.98)	4.92	59.36	0.09	0.12	0.12
5–10	3	1.20 (1.06–1.36)	1.27	0.00	0.53	0.60	0.60
≤10	20	1.06 (0.93–1.21)	16.92	82.27	0.001	0.46	0.50
>10	17	0.99 (0.85–1.16)	12.97	76.87	0.005	0.94	0.50
Group of indicators							
7 indicators ^1^	23	1.04 (0.95–1.14)	64.89	66.1	0.00	0.75	0.96
4 indicators ^2^	14	1.12 (0.96–1.30)	55.52	76.58	0.00	0.39	0.70
AA_Hannum	3	0.93 (0.74–1.16)	4.19	52.28	0.12	0.39	0.60
AA_Horvath	3	0.95 (0.86–1.05)	1.97	0.00	0.37	0.29	0.60
AA_Pheno	5	1.18 (0.95–1.48)	12.10	66.94	0.017	0.59	0.33
AA_Grim	3	1.97 (1.57–2.47)	1.14	0.00	0.57	0.62	0.60
3 indicators ^3^	8	0.96 (0.89–1.04)	4.05	0.00	0.77	0.026	0.14
IEAA_Hannum	3	0.95 (0.79–1.14)	2.83	29.28	0.24	0.19	0.12
IEAA_Horvath	3	0.97 (0.87–1.08)	0.58	0.00	0.75	0.046	0.12
Type of indicators							
Hannum (AA + IEAA)	6	0.95 (0.85–1.08)	7.02	28.77	0.22	0.023	0.19
Horvath (AA + IEAA)	6	0.96 (0.89–1.03)	2.64	0.00	0.76	0.023	0.13
Pheno (AA + IEAA)	7	1.09 (0.91–1.31)	16.73	64.13	0.010	0.68	0.88
Grim (AA)	3	1.97 (1.57–2.47)	1.14	0.00	0.57	0.62	0.60

^1^ AA_Hannum, AA_Horvath, AA_Grim, AA_Pheno, IEAA_Hannum, IEAA_Horvath, IEAA_Pheno. ^2^ AA_Hannum, AA_Horvath, AA_Grim, AA_Pheno. ^3^ IEAA_Hannum, IEAA_Horvath, IEAA_Pheno. ^a^ Risk estimates were calculated using the random effects model. ^b^ Number of data used to calculate the risk.

## Data Availability

The data presented in this study are available upon request from the corresponding author.

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
