# Peer review of "Methylation Biomarkers of Lung Cancer Risk: A Systematic Review and Meta-Analysis"

_cancers, 2025, doi:10.3390/cancers17040690_

Round 1

Reviewer 1 Report

Comments and Suggestions for Authors

Methylation Biomarkers in Lung Cancer Risk: A Systematic Review and Meta-Analysis

Lung cancer is the leading cause of cancer death globally among both men and women. DNA methylation (DNAm) patterns in blood, tumor tissue, or other bodily fluids are being discovered as potential biomarkers for early detection, prognosis, and monitoring of treatment response in lung cancer. Studies have shown that certain genes, such as CDKN2A, TP53, and RASSF1A, exhibit frequent methylation in lung cancer, and this can be detected in plasma or serum samples. So, DNAm serves as both a mechanistic driver of lung cancer and a promising biomarker for its detection and monitoring.

This review and meta-analysis investigated the association between DNA methylation (DNAm) in blood and the risk of developing lung cancer (LC). By synthesizing data from existing studies, the authors found that higher levels of DNAm are correlated with an increased risk of LC. The review addresses an important topic and provides a fairly comprehensive analysis of the link between DNAm in blood and lung cancer risk. Though the review is clear but could be improved with additional information and revision. For instance, the supplementary data was not provided, so the conclusions drawn either were not supported by the listed citations or the authors’ own data. Some results require further explanation, and there are instances where references are repeated in the bibliography. Additionally, specific statements in the review appear to contradict the authors' own assertions, warranting careful reconsideration. The formatting and structure of the document also need attention, with some sections and references requiring revision. Finally, the statistical analysis should be rigorously re-checked to ensure its accuracy.

Line 26-27: “Searches were conducted in PubMed, Web of Science, and Scopus for studies published until February 2024The statement contradicts the authors own assertion in line 261-261 that “Studies have been published from 2015 to 2023” This needs to be clarified?

Line 30: “DNAm levels and LC risk (OR 1.24, 95% CI 1.10-1.39; I² = 71.31%, p = 0.0001).” The highlighted values don’t follow the ones provided in table 2. Could you please justify?

Line 34: “No publication bias was found.” The statement contradicts the authors own assertion made in line 329 that “Publication bias was detected for ALL….” Could you please justify your statement.

Line 58: “that LC can be found also in people without a history of high exposure risk such as smok-ing[11]” Need to revise the sentence.

Line 90: “and PRISMA statement” Could you please expand on term PRISMA

Line 103: “database are shown in Table S1” Table S1 not shown

Line 124: “The effect estimates reflect a unitless increase in DNAm,” The statement is unclear.

Line 128: “as shown in Table S2 and Table S3.” Table S2 and Table S3 are not shown

Line 127: “The Newcastle–Ottawa Scale (NOS) was used to assess the quality of the literature”. Reference cited doesn’t provide any clue about NOS, need to correct it.

Line 221-226:  Could you provide ‘n’ individually for each report excluded (i.e. for Single gene methylation, Animal model, Promoter methylation and Mendelian Randomization) in the figure and provide details with their reasons in a supplementary table.

Line 239: “since they were studies on single gene methylation,” the statement is unclear, need revision.

Line 240-241: “11 studies were eligible for inclusion in the systematic review and meta-analysis [43], [44], [45], [46], [47], 230 [48], [49], [50], [51], [52], [53].” The studies cited do not follow the Table 1 citations.

Line 247-248: Table 1: The studies cited don’t follow the ones that got eligible at the end of selection process. Some of the columns in the table are emptied and information is missing such as Race (% white). Table needs formatting too

Line 251: “ESTHER, Epidemiologische Studie zu Chancen der Verhütung, Früherkennung und 251 optimierten Therapie chronischer Erkrankungen in der älteren Bevölkerung;Needs Correction

Line 259: “New Castel Ottawa Scale” Needs correction

Line 278-279: “All DNAm measure definitions, which are taken in consideration, are described in 278 Table S4.” Table S4 not shown

Line 279: “Regarding single epigenetic clock, we observed a statical association only for” Needs correction.

Line 313: Would be great if the scale for statistics is displayed in the forest plot.

Line 279-280: “Regarding single epigenetic clock, we observed a statical association only for 279 epigenetic age acceleration of GrimAge (OR=1.97; 95% C.I. 1.57-2.47).” Could you please refer to which table/figure has the data been shown?

Line 327: “… to 1.17 (95% CI 1.10–1.24, p < 0.001), omitting the study of Michaud et al. [53].” It’s not clear which figure or table this is pointed to? The data not shown.

Line 328: Could you please expand “mtDNAcn”?

Line 330: Could you please explain the results of funnel plot for publication Bias mitochondrial DNA copy number?

Line 333

Line 385: “CDKN2A that has been individuated among the first possible new biomarkers of early LC.” The statement is unclear, please do revise. Expand “CDKN2A” i.e. cyclin dependent kinase inhibitor 2A.

Line 385: “CDKN2A that has been individuated among the first possible new biomarkers of early LC detection[55],” Incorrect reference cited.

Line 386-387:  “..by a wider point of view as offered by many more modern measures and indicators such as the epigenetics clocks that can also relate to LC[56].” The statement is speculative and not supported by the cited reference. Please justify? Reference 56: “The findings from their study did not support a positive association between three different biological age acceleration measures and risk of lung cancer, so the association of epigenetic age with lung cancer in never smokers was not found.”

Line 388-389: “strong significant association for the overall effect and for several types of stratification, including smoking status.” Are the results shown? Could you please mention the figure number or table?

Line 391-394: “Interestingly, the follow-up period seems to represent an important …… ten years.,” Could you please mention/cite the table or figure these statements refer to?

Line 395: “It could be hypnotized that epigenetic modifications…” The statement is unclear.

Line397-400: This is in accordance with some studies that observed time since quitting smoking appears to be associated with greater reversal of DNAm patterns to non-smoking levels, although most dynamic epigenetic changes reverse within 5 years of smoking cessation[57].” The statement could be revised to make it clearer. While some methylation changes induced by smoking do indeed reverse with time after quitting, ‘complete reversal’ may not always occur, and some changes could remain for a long time. The reference cited suggests that smoking leads to changes in DNA methylation that can last even after someone stops smoking, but these changes may gradually decrease or become less significant over time. It highlights the lasting impact of smoking on the genome and the potential for some degree of recovery after cessation, though the data did confirm that within 5 years of smoking cessation there was a highest return of DNA methylation level.

Line 408-413: “the authors adjusted these clocks by regress…… then resulting in more comparable indicators”. It’s a bit of a long sentence and unclear. Please revise it.

Line 438: “the Newcastle-Ottawa Scale (NOS) ensuring” May not need to expand the term ‘NOS’ if already been stated earlier.

Line 440: “NOS Scale”. Since ‘S’ in NOS refers to Scale, so no need to repeat it.

Line 510: Reference needs correction

Line 596-597 and Line 620-621: Reference 51 and 63 are duplicate references

Line 624-627: References 65 and 66 are duplicate references

Line 600-601 and 606-607: Reference 53 and 56 are duplicate references

Line 533-534 and line 618-619: Reference 21and 62 are duplicate references

Comments on the Quality of English Language

Quality could be improved, some of the comment listed above.

Author Response

Line 26-27: “Searches were conducted in PubMed, Web of Science, and Scopus for studies published until February 2024 “The statement contradicts the authors own assertion in line 261-261 that “Studies have been published from 2015 to 2023” This needs to be clarified?

We would like to clarify the references made in lines 26-27 and 261 of our manuscript.

In lines 26-27, we refer to the research conducted using three search databases, including all articles published up to February 2024.

In line 261 we refer specifically to the articles included in the systematic review and meta-analysis after the first records screening and selection obtaining that the first included study was published in 2015 (Levine ME et al.), and the most recent ones were published in 2023 (Michaud DS et al. and Dogue PA et al.).

During the records screened for title and abstract revision phase, 1,546 records were excluded after reviewing their titles and abstracts as reported in the flowchart and the "Study Selection" section.

Line 30: “DNAm levels and LC risk (OR 1.24, 95% CI 1.10-1.39; I² = 71.31%, p = 0.0001).” The highlighted values don’t follow the ones provided in table 2. Could you please justify?

We thank the reviewer for pointing this out. We revised and there was a typo in the manuscript. The correct value is I²: 93.90, as also correctly reported in Section 3.3 Meta-Analysis DNAm. We have substituted the correct value in the Abstract.

Line 34: “No publication bias was found.” The statement contradicts the authors own assertion made in line 329 that “Publication bias was detected for ALL….” Could you please justify your statement.

We thank the reviewer for pointing this out. There was a typo in the abstract and we recognized the inconsistency. The corrected statement aligns with line 329, where a publication bias was detected.

Line 58: “that LC can be found also in people without a history of high exposure risk such as smok-ing[11]” Need to revise the sentence.

We thank the reviewer for pointing this out. We have revised and changed sentences with the following:

It is well known that LC can also occur in subjects without a history of high-risk exposure, such as smoking [11]. However, these populations are often not effectively reached by conventional prevention campaigns.

Line 90: “and PRISMA statement” Could you please expand on term PRISMA

We agree and we have expanded the term PRISMA as follows:

“(Preferred Reporting Items for Systematic reviews and Meta-Analyses)”

Line 103: “database are shown in Table S1” Table S1 not shown

Thank you for your valuable feedback. Your comment regarding the missing supplementary files made us question where they were. We initially believed they had been uploaded to the platform as part of a ZIP file. In fact, we have assigned names to the supplementary files in the “Supplementary Files” section before the References

After contacting the editors of the journals, they confirmed that they had uploaded the supplementary files on 28 Jenuary 2025.

Line 124: “The effect estimates reflect a unitless increase in DNAm,” The statement is unclear.

We agree with the reviewer and we have changed the statement to make it clearer as following:

The effect estimates represent a relative change in DNAm rather than an absolute in-crease measured in specific units, which may be associated with aging as measured by epigenetic clocks, smoking, or the methylation risk score.”

Line 128: “as shown in Table S2 and Table S3.” Table S2 and Table S3 are not shown

Thank you for your valuable feedback. Your comment regarding the missing supplementary files made us question where they were. We initially believed they had been uploaded to the platform as part of a ZIP file. In fact, we have assigned names to the supplementary files in the “Supplementary Files” section before the References.

After contacting the editors of the journals, they confirmed that they had uploaded the supplementary files on 28 Jenuary 2025.

Line 127: “The Newcastle–Ottawa Scale (NOS) was used to assess the quality of the literature”. Reference cited doesn’t provide any clue about NOS, need to correct it.

We have added the correct reference, which states that the Newcastle-Ottawa Scale (NOS) is used to assess the quality of the literature, as follows:

[32] Ottawa Hospital Research Institute. (n.d.). Retrieved January 28, 2025, from https://www.ohri.ca/programs/clinical_epidemiology/oxford.asp

Line 221-226:  Could you provide ‘n’ individually for each report excluded (i.e. for Single gene methylation, Animal model, Promoter methylation and Mendelian Randomization) in the figure and provide details with their reasons in a supplementary table.

As requested, we have added the ‘n’ for each group in Figure 1.

We have provided additional details and clarified the reasons for the points in the figure into a supplementary table as follows:

Table S4. Reasons of the excluded studies

Reason of exclusion

Motivation

Single gene methylation

Methylation is typically studied at the single gene level. However, regression models for methylation generally consider whole genome activity, which is why we did not focus exclusively on single gene methylation.

Animal model

Our systematic review and meta-analysis are focused on human studies, and animal models were not included in our systematic review and meta-analysis.

Promoter methylation

Methylation as an epigenetic mechanism is not limited to the promoter region. It can also occur in other regions of the gene, which is why we did not limit our systematic review and meta-analysis to only promoter methylation.

Mendelian Randomization

The Newcastle-Ottawa Scale (NOS) cannot be applied to randomized studies like Mendelian randomization, so these articles were excluded from this systematic review and meta-analysis.

Line 239: “since they were studies on single gene methylation,” the statement is unclearneed revision.

To clarify the statement, we have added the following sentence:

We excluded articles that addressed these points from this systematic review and meta-analysis for the following reasons: methylation of a single gene was not the primary focus of methylation regression models, this systematic review and meta-analysis focused on human studies and not animal models, methylation as an epigenetic mechanism extends beyond the promoter region, and the Newcastle-Ottawa Scale is not applicable to randomized studies such as Mendelian Randomization.”

Line 240-241: “11 studies were eligible for inclusion in the systematic review and meta-analysis [43], [44], [45], [46], [47], 230 [48], [49], [50], [51], [52], [53].” The studies cited do not follow the Table 1 citations.

We thank the reviewer for pointing this out. We agree the correct references are those listed in Figure 1. The initial (numbers) in Table 1 were markers that we had assigned to help us to identify the corresponding articles.

We have now updated the references, replacing the initial (numbers) with the correct [references].

Line 247-248: Table 1: The studies cited don’t follow the ones that got eligible at the end of selection process. Some of the columns in the table are emptied and information is missing such as Race (% white). Table needs formatting too

Thank you for your valuable feedback.

Regarding references, we corrected them by replacing the initial (marker numbers) with the correct [references].

Regarding the Race column (% white), it should remain in the table because Zhao N et al. 2021 [48] is the only study that includes this characteristic in the description of case and control populations.

Regarding blank columns, these fields are blank because the corresponding information is not given in the respective articles. To clarify this concept, we have added NA (not available) and added a specific annotation in the description of the table.

Line 251: “ESTHER, Epidemiologische Studie zu Chancen der Verhütung, Früherkennung und 251 optimierten Therapie chronischer Erkrankungen in der älteren Bevölkerung;” Needs Correction

We thank the reviewer for pointing this out. We have changed with the corresponding acronym in english as follow:

Epidemiological Study on the Chances of Cure, Early Detection and Optimized Therapy of Chronic Diseases in the Elderly Population”

Line 259: “New Castel Ottawa Scale” Needs correction

We agree with the reviewer. We have corrected as follow:

Newcastle-Ottawa Scale”

Line 278-279: “All DNAm measure definitions, which are taken in consideration, are described in 278 Table S4.” Table S4 not shown

Thank you for your valuable feedback. Your comment regarding the missing supplementary files made us question where they were. We initially believed they had been uploaded to the platform as part of a ZIP file. In fact, we have assigned names to the supplementary files in the “Supplementary Files” section before the References.

After contacting the editors of the journals, they confirmed that they had uploaded the supplementary files on 28 Jenuary 2025.

Line 279: “Regarding single epigenetic clock, we observed a statical association only for” Needs correction.

We have corrected the sentence as follow:

“Regarding the single epigenetic clock (Table 2), we observed a statistical association only for ” 

Line 313: Would be great if the scale for statistics is displayed in the forest plot.

We thank the reviewer. We have added the scale for statistics in the forest plot.

Line 279-280: “Regarding single epigenetic clock, we observed a statical association only for 279 epigenetic age acceleration of GrimAge (OR=1.97; 95% C.I. 1.57-2.47).” Could you please refer to which table/figure has the data been shown?

We thank the reviewer. We have added the reference table

“(Table 2)”

Line 327: “… to 1.17 (95% CI 1.10–1.24, p < 0.001), omitting the study of Michaud et al. [53].” It’s not clear which figure or table this is pointed to? The data not shown.

Thank you for your comment. In the manuscript, we conducted sensitivity analyses to evaluate the robustness of our findings by sequentially omitting each study and recalculating the pooled risk estimates. As highlighted in the text, these analyses demonstrated that no single study disproportionately influenced the overall estimates. To ensure the manuscript remains concise and focused, we opted to summarize the key results of the sensitivity analyses directly in the text rather than generating additional forest plots for each omission.

Given the consistency of the estimates, we believe that providing these summary statistics (e.g., the range of risk estimates) is sufficient to illustrate the stability of our findings without the need for additional figures or tables.

Line 328: Could you please expand “mtDNAcn”?

We thank the reviewer for pointing this out. We have corrected with the right title:

Publication Bias DNAm”

Line 330: Could you please explain the results of funnel plot for publication Bias mitochondrial DNA copy number?

We thank the reviewer for this valuable observation. There was a typo and the analysis refers to DNAm (DNA methylation), not mitochondrial DNA copy number (mtDNAcn). This error occurred because, at the beginning of the systematic review and meta-analysis, the scope was to include mtDNAcn, but in the end the focus was on DNAm.

Line 333

Line 385: “CDKN2A that has been individuated among the first possible new biomarkers of early LC.” The statement is unclear, please do revise. Expand “CDKN2A” i.e. cyclin dependent kinase inhibitor 2A.

We expanded “CDKN2A” and we clarify the statement as follow:

with a particular focus on specific regions, such as the promoter region of the CDKN2A gene (Cyclin Dependent Kinase Inhibitor 2A), which has been identified as one of the first potential new biomarkers for the early detection of LC [56]. In addition, a more comprehensive approach has been taken through modern measures, such as epigenetic clocks, which can also be related to LC.”

Line 385: “CDKN2A that has been individuated among the first possible new biomarkers of early LC detection [55],” Incorrect reference cited.

Thank you for your comment. We would like to clarify that the reference cited is correct. Reviewing the article, the authors in the introduction also explain the silencing of CDKN2A, which is consistent with our statement. As mentioned in the article, CDKN2A has been identified as one of the early biomarkers for lung cancer diagnosis, as follows:

“Importantly, silencing of genes such as CDKN2A (p16), O6-methylguanine-DNA methyltransferase (MGMT), and adenomatous polyposis coli (APC) is detected in alveolar and bronchial epithelium of smokers, in precursor lesions to adenocarcinoma and squamous cell carcinoma, and the prevalence of gene methylation increases during disease progression”

Line 386-387:  “..by a wider point of view as offered by many more modern measures and indicators such as the epigenetics clocks that can also relate to LC[56].” The statement is speculative and not supported by the cited reference. Please justify? Reference 56: “The findings from their study did not support a positive association between three different biological age acceleration measures and risk of lung cancer, so the association of epigenetic age with lung cancer in never smokers was not found.”

Thank you for your valuable comment. We recognize that the reference we initially cited ([56]) does not appropriately support our statement. To address this, we have revised the sentence and replaced the reference with studies that specifically investigate the relationship between epigenetic clocks and lung cancer ([45], [51]). Additionally, we have also highlighted studies that found no association between epigenetic clocks and lung cancer ([46], [54]).

Line 388-389: “strong significant association for the overall effect and for several types of stratification, including smoking status.” Are the results shown? Could you please mention the figure number or table?

Thank you for your comment. We have mentioned the correspondent Table (Table 2) as follow:

We found a strong significant association for the overall effect and for several types of stratification, including smoking status (Table 2).”

Line 391-394: “Interestingly, the follow-up period seems to represent an important …… ten years.,” Could you please mention/cite the table or figure these statements refer to?

Thank you for your comment. We have mentioned the correspondent Table (Table 2) as follow:

Interestingly, the follow-up period (Table 2) seems to represent an important discrimination in terms of DNAm biomarkers and LC risk: for time periods ≤5 years and from 5 to 10 years of observation, the risk increases in a statistically significant manner with a higher risk in the  ≤5 years class.”

Line 395: “It could be hypnotized that epigenetic modifications…” The statement is unclear.

we have clarified the statement as follow:

It can be hypnotized that epigenetic modifications have a much higher impact on LC risk within the first five years of their occurrence, and their effect gradually decreases over time

Line397-400: “This is in accordance with some studies that observed time since quitting smoking appears to be associated with greater reversal of DNAm patterns to non-smoking levels, although most dynamic epigenetic changes reverse within 5 years of smoking cessation[57].” The statement could be revised to make it clearer. While some methylation changes induced by smoking do indeed reverse with time after quitting, ‘complete reversal’ may not always occur, and some changes could remain for a long time. The reference cited suggests that smoking leads to changes in DNA methylation that can last even after someone stops smoking, but these changes may gradually decrease or become less significant over time. It highlights the lasting impact of smoking on the genome and the potential for some degree of recovery after cessation, though the data did confirm that within 5 years of smoking cessation there was a highest return of DNA methylation level.

Thank you for your thoughtful comment. Upon reviewing the referenced study, we recognize that while some methylated CpG sites return to levels observed in never smokers within five years of smoking cessation, a subset of CpGs remains altered even after 30 years. To better reflect this finding, we have revised the sentence following your considerations:

“This is in accordance with some studies that observed that some methylation changes induced by smoking can reverse within 5 years of quitting. Anyway, a large portion of CpG sites remain persistently altered for decades. These findings highlight both the substantial reversibility of smoking-related epigenetic modifications and the potential for long-lasting effects at specific loci”

Line 408-413: “the authors adjusted these clocks by regress…… then resulting in more comparable indicators”. It’s a bit of a long sentence and unclear. Please revise it.

We agree with the reviewer. We have revised the sentence as follow:

The authors adjusted these clocks by regressing DNAm age on chronological age and calculating the difference between the observed chronological age and the fitted DNAm age, obtaining DNAm age acceleration (AA) [53]. Intrinsic epigenetic age acceleration (IEAA) was then derived using the residuals from the linear regression of DNAm age on chronological age, further adjusted for estimated blood cell composition [53]. These adjustments allowed for more comparable indicators.

Line 438: “the Newcastle-Ottawa Scale (NOS) ensuring” May not need to expand the term ‘NOS’ if already been stated earlier.

We thank and agree with the reviewer.

Line 440: “NOS Scale”. Since ‘S’ in NOS refers to Scale, so no need to repeat it.

We thank and agree with the reviewer.

Line 510: Reference needs correction

Line 596-597 and Line 620-621: Reference 51 and 63 are duplicate references

We thank the reviewer for pointing this out. We eliminated one of the duplicate references

Line 624-627: References 65 and 66 are duplicate references

We thank the reviewer for pointing this out. We eliminated one of the duplicate references

Line 600-601 and 606-607: Reference 53 and 56 are duplicate references

We thank the reviewer for pointing this out. We eliminated one of the duplicate references

Line 533-534 and line 618-619: Reference 21and 62 are duplicate references

We thank the reviewer for pointing this out. We eliminated one of the duplicate references

Reviewer 2 Report

Comments and Suggestions for Authors

The authors have utilized the term 'lung cancer risk" in the title and throughout the paper.  It is not clear if this review is about lung cancer risk or diagnosis.  This needs to be clarified. 

Author Response

The authors have utilized the term 'lung cancer risk" in the title and throughout the paper.  It is not clear if this review is about lung cancer risk or diagnosis.  This needs to be clarified. 

We thank the reviewer, and we better clarified this aspect in the main text in the Data extraction and Quality Assessment paragraph as follows:

“Our results are derived from the analysis of these regressor models constructed in studies that compared DNA methylation levels between populations diagnosed with lung cancer and control groups without lung cancer. While these studies used diagnosis as a reference for model construction, the resulting regression model is capable of assessing risk based on given methylation levels”.

Reviewer 3 Report

Comments and Suggestions for Authors

None

Author Response

Thank you for your appreciation and support.

Reviewer 4 Report

Comments and Suggestions for Authors

This study examines the usefulness of DNA methylation (DNAm) biomarkers in assessing lung cancer risk through a systematic review and meta-analysis, and provides important findings for early cancer detection and prevention strategies. The authors conducted a literature search in multiple databases and followed established guidelines such as PRISMA and MOOSE.  Here are some comments that need to be considered.

1.  The high I² value of over 70% in the overall analysis, including case-control studies, indicates a high degree of heterogeneity. Although this is recognised and addressed via subgroup analyses, it limits the generalisability of the findings.. In principle, meta-analysis should not be conducted if there is a high degree of heterogeneity between the studies. In such cases, it is better to limit the analysis to a systematic review only.

2. The lack of a statistically clear dose-response relationship between methylation levels and lung cancer risk raises questions about the interpretation of the results. It would be desirable to add more studies that include quartile data on methylation levels to clarify the dose-response relationship, or present a reasonable argument in discussion part. Changes in DNA methylation caused by smoking usually play a role in regulating gene expression, so long-term smoking can cause abnormalities in gene expression, which may increase the risk of cancer. (Roby Joehanes, et al. Circ Cardiovasc Genet. 2016) Specifically, smoking can promote methylation in some genes, which can lead to the suppression of tumour suppressor genes and the activation of oncogenes.The longer a person smokes, the more these epigenetic changes accumulate and the stronger their impact becomes, so the risk of lung cancer tends to increase.

Author Response

This study examines the usefulness of DNA methylation (DNAm) biomarkers in assessing lung cancer risk through a systematic review and meta-analysis, and provides important findings for early cancer detection and prevention strategies. The authors conducted a literature search in multiple databases and followed established guidelines such as PRISMA and MOOSE.  Here are some comments that need to be considered.

  1. The high I² value of over 70% in the overall analysis, including case-control studies, indicates a high degree of heterogeneity. Although this is recognized and addressed via subgroup analyses, it limits the generalizability of the findings. In principle, meta-analysis should not be conducted if there is a high degree of heterogeneity between the studies. In such cases, it is better to limit the analysis to a systematic review only.

We thank the reviewer for pointing this out. We agree that there is an high degree of heterogeneity (I² > 70%) observed in our meta-analysis, particularly due to the inclusion of case-control studies. As discussed in the Limitations and Strengths paragraph, this heterogeneity can be attributed to differences in several analytical factors, including the various types of epigenetic clocks and indicators used in the selected studies.

To partially solve this problem, we conducted subgroup analyses to explore potential sources of variability and improve the interpretability of our results as well as including additional sensitivity analyses. Moreover, while high heterogeneity can be a limitation, it does not automatically preclude the use of meta-analysis, especially in fields where variation between studies is expected due to differences in study design, population characteristics, or methodologies. By synthesizing the available evidence, we provide a more comprehensive and quantitative assessment than a systematic review alone would allow.

  1. The lack of a statistically clear dose-response relationship between methylation levels and lung cancer risk raises questions about the interpretation of the results. It would be desirable to add more studies that include quartile data on methylation levels to clarify the dose-response relationship, or present a reasonable argument in discussion part. Changes in DNA methylation caused by smoking usually play a role in regulating gene expression, so long-term smoking can cause abnormalities in gene expression, which may increase the risk of cancer. (Roby Joehanes, et al. Circ Cardiovasc Genet. 2016) Specifically, smoking can promote methylation in some genes, which can lead to the suppression of tumour suppressor genes and the activation of oncogenes.The longer a person smokes, the more these epigenetic changes accumulate and the stronger their impact becomes, so the risk of lung cancer tends to increase.

We appreciate the reviewer's insightful comment. We acknowledge that our meta-analysis does not show a statistically clear dose-response relationship between methylation levels and lung cancer risk. However, this does not necessarily imply the absence of a biological association. As suggested, we have expanded the discussion by incorporating evidence from Joehanes et al. (2016) to provide a mechanistic explanation for the potential long-term effects of smoking-induced DNA methylation changes in the discussion section as follows:

From a biological point of view, smoking is known to promote DNA methylation at certain loci, leading to the silencing of tumor suppressor genes and the activation of oncogenes. Over time, these epigenetic alterations can accumulate, potentially increasing the risk of lung cancer even in the absence of a direct dose-response pattern in the available data. This is consistent with epidemiological findings showing that the longer a person smokes, the higher their lung cancer risk. Furthermore, the complexity of epigenetic regulation means that methylation changes may have non-linear effects on gene expression, making it challenging to detect a straightforward dose-response trend in case-control studies. Other confounding factors, such as inter-individual variability in epigenetic plasticity and the interplay with additional genetic and environmental influences, may further obscure a simple dose-response relationship [63].

Round 2

Reviewer 3 Report

Comments and Suggestions for Authors

None

Reviewer 4 Report

Comments and Suggestions for Authors

Thank you for the response.  I recommend the paper be accepted.